# Nitropyridine-1-Oxides as Excellent π-Hole Donors: Interplay between σ-Hole (Halogen, Hydrogen, Triel, and Coordination Bonds) and π-Hole Interactions

**DOI:** 10.3390/ijms20143440

**Published:** 2019-07-12

**Authors:** Bartomeu Galmés, Antonio Franconetti, Antonio Frontera

**Affiliations:** Department of Chemistry, Universitat de les Illes Balears, Crta de Valldemossa km 7.5, 07122 Palma de Mallorca (Baleares), Spain

**Keywords:** CSD analysis, π-hole interactions, σ-hole interactions, supramolecular chemistry, cooperativity

## Abstract

In this manuscript, we use the primary source of geometrical information, i.e., Cambridge Structural Database (CSD), combined with density functional theory (DFT) calculations (PBE0-D3/def2-TZVP level of theory) to demonstrate the relevance of π-hole interactions in para-nitro substituted pyridine-1-oxides. More importantly, we show that the molecular electrostatic potential (MEP) value above and below the π–hole of the nitro group is largely influenced by the participation of the N-oxide group in several interactions like hydrogen-bonding (HB) halogen-bonding (XB), triel bonding (TrB), and finally, coordination-bonding (CB) (N^+^–O^−^ coordinated to a transition metal). The CSD search discloses that *p*-nitro-pyridine-1-oxide derivatives have a strong propensity to participate in π-hole interactions via the nitro group and, concurrently, N-oxide group participates in a series of interactions as electron donor. Remarkably, the DFT calculations show from strong to moderate cooperativity effects between π–hole and HB/XB/TrB/CB interactions (σ-bonding). The synergistic effects between π-hole and σ-hole bonding interactions are studied in terms of cooperativity energies, using MEP surface analysis and the Bader’s quantum theory of atoms in molecules (QTAIM).

## 1. Introduction

Chemists working in supramolecular chemistry, crystal engineering, and materials science [1,2,3] frequently put their faith in traditional hydrogen bonding (HB) [4,5,6], strong donor–acceptor π-stacking interactions and, to a lesser extent, halogen bonding (XB) interactions [7,8,9,10,11,12,13,14]. However, other noncovalent σ/π-hole interactions involving *p*-block elements have undertaken a fascinating progress [15,16,17,18,19,20,21,22,23,24,25]. For instance, the strength and directionality of chalcogen, pnictogen, tetrel, and aerogen bonds are comparable to XB [26,27,28,29,30,31,32,33,34,35,36,37,38,39,40,41,42]. Striking and distinctive features of σ–hole interactions involving Groups 14-18 of elements with respect to HBs are the greater hydrophobicity and directionality. These features have been recently used to design catalysts that function in apolar media. For instance, benzodiselenazole and tris(perfluorophenyl)arsane have been used for chalcogen and pnictogen bonding based catalysis that takes advantage of σ–holes on selenium/arsenic atoms pointing to the catalytic site [43,44,45,46,47].

Apart from the emerging σ-hole interactions, noncovalent interactions involving π-holes are also gaining attention. They have been studied in trivalent boron and aluminium compounds, carbonyls, and particularly in nitro-derivatives and related compounds [48,49,50,51,52,53,54]. Nitrocompounds are common, easy to synthesize, and a great deal of information is accessible in the largest source of geometrical information: the Cambridge Structural Database (CSD) [55]. Additionally, in the gas-phase, it has been experimentally demonstrated by rotational spectroscopy that a noncovalent π–hole complex is formed by the interaction of the free lone pair of trimethylamine and nitroethane. Remarkably, this study shows that the Me_3_N···NO_2_Me π-hole interaction dominates over H-bonding [56]. Furthermore, the 3D framework observed in the solid state X-ray structure of 2,4-dinitro-2,4-hexadiene is generated exclusively by the formation of short nitro···nitro π–hole contacts involving the O-atoms as lone pair donors and the N-atom (π-hole) as electron acceptor [57].

Molecular recognition and self-assembly processes [58,59] are often possible due to synergetic effects between several intermolecular interactions, thus having many implications in biochemistry, crystal engineering, and material science [60,61]. In fact, the ability of noncovalent forces to control and tune highly specific binding is often due to an intricate combination of noncovalent forces acting cooperatively [62]. The understanding of these effects can be obtained from X-ray crystal structures (analyzing the CSD) and quantum chemical calculations, thus providing useful geometric and energetic information.

We and others have demonstrated both experimentally and theoretically important cooperativity effects in complexes where two or more noncovalent interactions coexist [63,64,65,66,67,68]. Moreover, we have recently analyzed cooperativity effects between π-hole and halogen bonding interactions in 4-nitropyridine and 4-cyano-nitrobenzene [69], without experimental support from the CSD. In this new work, we report a combined theoretical density functional theory (DFT) and CSD study where the interplay between π–hole and several σ-hole interactions are analyzed. The π–hole interaction involves the N-atom of the nitro group of *p*-nitro-pyridine-1-oxide and the other interactions involve the N-oxide group (as electron donor). We have carefully chosen *p*-nitro-pyridine-1-oxide (see Figure 1) because it contains an exocyclic O-atom suitable for interacting with Lewis acids and a π-hole donor nitro group adequate for the interaction with electron rich atoms. As detailed in Figure 1a, compound **1** has two π-holes (+22.5 and +19.5 kcal mol^−1^ at the PBE0-D3/def2-TZVP level of theory) which are adequate for interacting with Lewis bases. The molecular electrostatic potential (MEP) minimum is located at the O-atom of the N-oxide group (−27.0 kcal mol^−1^). In fact, the X-ray structure of compound **1** (Figure 1b) provides strong experimental support to the ability of nitro’s π-hole to interact with Lewis bases. The O-atom of the N-oxide group is situated above the C–N bond exactly over the location of the π-hole, with the O···N distance slightly shorter than the O···C distance.

As starting point, in this manuscript we have analyzed the CSD in order to investigate the existence of π-hole interactions in *p*-nitropyridine-N-oxide derivatives in crystal structures. In addition, we demonstrate the existence of ternary complexes where the N^+^–O^−^ group is engaged in other interactions simultaneously, particularly hydrogen-, halogen-, triel-bonding, and coordination bonds (HB, XB, TrB, and CB, respectively). Consequently, we have focused our DFT study to the computation of the geometric and energetic features of binary HB, XB, CB, and TrB-bonded complexes **2**–**5** and π–hole (πH) complexes **6**–**10** depicted in Scheme 1. Next, we have calculated the three component systems where πH and HB/XB/CB/TrB interactions exist in the same complex **11**–**30** represented in Scheme 2. These multicomponent systems allow us to study the mutual influence of both interactions. As electron rich atoms, we have used both anions and neutral lone pair donors, in order to investigate their effect in the cooperativity energies. In addition to the analysis of the energetic and geometric features in the ternary systems with respect to the binary complexes, we have also used the quantum theory of “atoms in molecules” (QTAIM) [70], to provide further insight into the synergistic effects. This method provides a useful criterion to define which atoms from donor and acceptor groups interact in a supramolecular complex and, also important, gives hints regarding the strength of the interaction [71,72]. Finally, the molecular electrostatic potential values and surfaces have been computed to investigate if the mutual reinforcement of the interactions is due to electrostatic effects.

## 2. Results and Discussion

### 2.1. CSD Search

We have started this investigation by doing a CSD search, since this database is a huge reservoir of geometrical information and frequently reveals details that have not been reported in the original work. For this search, we have simply used as quest *p*-nitropyridine-1-oxide and restricted the search to those structures determined by single crystal X-ray spectroscopy, with “no error” in the structure, 3D coordinates determined and no disorder. As a result, we have found 116 X-ray structures and co-crystals having the *p*-nitropyridine-1-oxide core. A manual inspection of the structures reveals that in most of them (80%) the nitro group participates in π-hole interactions with a variety of electron rich atoms (from Lewis bases to anions). Also remarkably, in those structures where a clear π-hole interaction is not observed (23 out of 116), the nitro group is stacked over an aromatic ring, thus interacting with the electron rich π-cloud. Therefore, this preliminary analysis confirms the strong ability of the nitro group to act as Lewis acid. Even more importantly, the analysis of the solid state architecture of the X-ray structures of *p*-nitropyridine-1-oxide derivatives also reveals that the N-oxide group participates in covalent/noncovalent interactions, as shown in Figure 2. For instance, the bis(4-nitropyridine-1-oxide)trans-dichloro-diaqua-copper(II) (refcode NPYOCU [72], see Figure 2a) structure forms dimers in the solid state where the O-atom of the nitro group is located over the N atom of the nitro group of the adjacent molecule at a distance that is significantly shorter than the sum of van der Waals radii (ΣR_vdW_ = 3.10 Å). The coordination of the N^+^–O^−^ group to the Cu(II) metal center is likely increasing the π–acidity of the nitro group, thus favouring the π-hole interaction. Similarly, in the (4-nitropyridine-1-oxide)-(trifluoroborane) adduct (refcode MUFZUJ [73], see Figure 2b) the N-oxide group is bonded to the BF_3_ molecule, increasing the intensity of the nitro’s π-hole that establishes a short contact with the O-atom of the adjacent molecule in the solid state. Figure 2c shows the self-assembled tetramer formed by four molecules of 3-iodo-2,6-dimethyl-4-nitropyridine-1-oxide (refcode XIHCOG [74]). A double π-hole interaction is established between the nitro group and, simultaneously, the N-oxide group forms a halogen bonding interaction with the iodine. A similar situation is observed in the cocrystal of 4-nitropyridine-1-oxide and 4-nitrophenol (refcode JUDNAX [75]), where the nitro group participates in a π-hole interaction with the nitro of the 4-nitrophenol and, concurrently, the N-oxide group establishes a hydrogen bonding interaction with the acidic H-atom of the 4-nitrophenol.

### 2.2. MEP Study

Figure 3 represents the MEP surfaces of binary complexes **2**–**5** plotted onto the van der Waals isosurface (0.001 a.u.) and Table 1 gathers the values of MEP at the π-hole (V_s,πh_), over the ring center (V_s,centroid_) and at the maximum (V_s,max_) that is located between the aromatic H-atoms in the ring plane (represented by an asterisk in Figure 3). The MEP value above the C–N bond of the nitro group is more positive than that over the center of the ring, thus revealing a slight preference for establishing π-hole instead of anion–π interactions in these complexes, at least from an electrostatic point of view. It is worthy to point out that the MEP values at the π-holes and at the maximum increase upon noncovalent complexation of **1** to either the HF or CF_3_I molecules. Interestingly, the increment is larger in any of both π-holes (over the ring center and over the C–NO_2_) than in the maximum (V_s,max_). Consequently, there is an enhanced electronic communication between the σ-hole interaction and the π-system. Therefore, a favorable interplay between either HB or XB and π-hole interactions is predictable by analyzing the MEP surfaces of the compounds. Regarding the covalent complexes **3** and **4**, the increment of MEP values at any of both π-holes and at the maximum is very significant. For instance, in the BF_3_ complex, the MEP value at the nitro’s π-hole increases from +22.5 in **1** to +43.5 kcal mol^−1^ in **5** and, similarly, over the ring centroid increases from +19.4 in **1** to +41.2 kcal mol^−1^ in **5**. The same behavior is observed for the AgCl complex. Therefore, the synergistic effects in the latter complexes are expected to be very important.

### 2.3. Two Component Complexes

The interaction energies (Δ*E*, kcal mol^−1^) and equilibrium distances (d, Å) of complexes **2**–**10** (see Scheme 1) are gathered in Table 2. Both energies and geometries were computed at the PBE0-D3/def2-TZVP level of theory. The interaction energies of complexes **2** and **3** are in the typical range of HB and XB interactions. The strength of the coordination complex **4** is stronger (–21.2 kcal mol^−1^) and finally the triel bonded complex **5** is very strong −73.1 kcal mol^−1^) due to the covalent nature of the O–B bond, as supported by the short distance (1.611 Å). The π-hole complexes with anionic donors (**6** and **7**) exhibit large interaction energies (>12 kcal·mol^−1^) and those with neutral Lewis basis are more modest, apart from the Me_3_N complex **8**.

Figure 4 shows the optimized geometries of the π-hole 1:1 complexes. The O/N lone pair donor atom in complexes **8** and **10** is located over the C–NO_2_ bond, displaced toward the N atom. In the anionic and nitromethane complexes the binding mode is ditopic with one atom pointing to the nitro group and the other to the ring centroid, so both π-holes are involved in the interaction. In these complexes, the nitro’s π-hole distance is longer than the anion–π distance (**6**, **7** and **9**). It is worthy to note that in the monotopic complexes **8** and **10**, the interaction is with the nitro’s π-hole, in good agreement with the MEP surface shown in Figure 1a. Finally, the π-hole equilibrium distances are shorter for the anionic complexes than for the neutral ones. They are in quite good agreement with the distances observed in the X-ray structures (see Figure 2).

### 2.4. Three Component Systems

The interaction energies (Δ*E*, kcal mol^−1^) and equilibrium distances (d, Å) of complexes of three component systems **11**–**30** (see Scheme 2) are summarized in Table 3. Remarkably, the equilibrium distances (d_πh_) of the π–hole interaction in the three component complexes **11**–**30** are shorter than in complexes **6**–**10** (Δd_πh_ values in binary complexes are negative,). Namely, the co-existence of the σ-bonding interaction strengthens the π–hole interaction. Similarly, the equilibrium distance of the σ-bonding interaction d_σB_ in the three component systems is also shorter compared to two component systems **2**–**5** (negative values in most of the cases for Δd_XB_ see Table 3). Consequently, the existence of the π–hole bonding reinforces the σ–bonding interaction. To make more evident the mutual influence between both interactions, we have selected complexes **16** and **28**, as illustrated in Figure 5. In complex **16**, the CF_3_I molecule interacts with N-oxide group with an equilibrium distance of 2.629 Å that represents a significant shortening of the σ–hole interaction (Δd_σh_ = −0.254 Å) with respect to complex **3** (see Table 2). Moreover, the three π-hole distances represented in the figure also shorten (Δd_XB_ = −0.016 Å for the O···N) with respect to complex **6** (see Figure 4a). This reveals that the mutual influence of both noncovalent interactions is communicated in both directions, from the π-hole to the orthogonal O-atom in the molecular plane and vice versa. In this particular complex, the σ–hole interaction is significantly more affected for the presence of the π-hole interaction than vice-versa, likely due to the anionic nature of the donor. In complex **28** (see Figure 5, right), the N···N and N···C equilibrium distances that characterize the π-hole interaction shorten with respect to the binary complex **8** (see Figure 4c) due to the presence of the BF_3_ Lewis acid connected to the N-oxide group. The shortening of the N···C distance is more pronounced than the N···N distance because the location of the π-hole over the C–N bond moves toward the carbon upon formation of the triel bond.

Table 3 also summarizes the cooperativity energies *E*_coop_ (see below Section 3), which give an approximation of the extra energetic stabilization that is gotten in the three component systems due to the interplay between σ- and π-hole interactions. We have computed these energies for complexes where both interactions are noncovalent (**11**−**20**). For the rest of complexes, we have computed the “binary” energies (Δ*E*_bin_) to evaluate cooperativity effects. The binary energies are computed by considering the ternary system as a binary one. That is, we have assumed that the either the coordination bond or the triel bond has been formed first and only evaluated the π-hole interaction. The different equations used are also illustrated in Figure 5. In the top of Figure 5 we show the equation used to compute the interaction energies (ΔE, second column of Table 3), see also computational methods. In addition, for complex **16** (both interactions are noncovalent), we have computed the cooperativity energies (*E*_coop_, third column of Table 3) using the equation shown at the bottom of Figure 5. Therefore, we have used the interaction energies of complexes **16** (Table 3), **6**, and **3** (Table 2) and an additional term [Δ*E*(AC)], which is the interaction energies of the nitrate (A) and CIF_3_ (C) as they stand in the ternary complex. In case of complex **16** the *E*_coop_ = –5.2 kcal mol^−1^ corresponds to the extra stabilization due to the coexistence of both interactions. For complex **28**, since one interaction is covalent (triel bond) we have computed Δ*E*_bin_, which is simply the interaction energy of **28** considering that the triel bond has been already formed. This interaction energy is –8.8 kcal mol^−1^, which is more negative than that of complex **8** (–7.0 kcal mol^−1^). The difference between both values is an estimation of the extra stabilization of the π-hole due to the presence of the triel bond. In all ternary complexes, the *E*_coop_ are negative and the ΔE_bin_ values are more negative than the corresponding binding energies of the binary complexes included in Table 2. This favorable synergistic effects are in agreement with the shortening of the equilibrium distances (apart from some exceptions commented below), see Δd_πh_ and Δd_σB_ values. For complexes **11**−**20** the computed *E*_coop_ values are larger for the anionic complexes than for neutral ones. This is also observed in complexes **21**−**30**, where the difference between ΔE_bin_ and ΔE values are larger for anions.

It is important to comment those particular cases where the π-hole equilibrium distance increases (instead of decreasing) in the three component systems in comparison to the relative two component dimers (positive Δd_πh_ values, see values in bold in Table 3). Apparently, it strongly disagrees with the favorable cooperativity energies obtained for these complexes. However, the enlargement of the C/O···N π–hole equilibrium distance is complemented by a shortening of the C/O···C distance because the electron rich atom gets closer to the ring center. Figure 6 depict two representative three component systems to exemplify this behavior. In complex **15** (Figure 6a) the C···N distances slightly enlarges (0.020 Å) and the C···C distance shortens (0.023Å) thus compensating for the apparent weakening of the interaction and resulting in favorable cooperativity due to the additional reinforcement of the H-bonding interaction. A similar behavior is observed in complex **19** (Figure 6b). That is, one π-hole distance increases (O···N) and two shorten (O···C and O···Cg) thus evidencing that the interaction of nitromethane with the π-system of 4-nitropyridine-1-oxide is reinforced, in agreement with the negative value of *E*_coop_.

As commented above, we have previously studied cooperativity effects between π-hole and halogen bonding interactions in p-nitropyridine and p-nitrobenzonitrile compounds [69]. In their ternary complexes with Lewis bases and halogen bond donors the reported cooperativity energies ranged from –4.6 kcal mol^−1^ for anionic donors and strong electron acceptors (CF_3_I) to –0.1 kcal mol^−1^ for weak donors (CO) and acceptors (CF_3_Cl). In case of nitropyridine-N-oxide studied in this work, the *E*_coop_ values are larger in absolute value ranging from –0.2 kcal mol^−1^ for weak electron donors to –9.1 kcal mol^−1^ for nitrate anion in combination with HF as H-bonding donor (complex **11**).

### 2.5. QTAIM Results

We carried out the QTAIM analysis of the two and three component systems studied on this work. The values of the electron charge density ρ(r) and its Laplacian [∇^2^ρ(r)] measured at the bond critical points (CPs) that emerge upon complexation give information on the strength of noncovalent interaction as previously shown in the literature. In fact, the values of ρ(r) have been used before to analyze cooperativity effects in several multicomponent systems [64,65,66,67,68,69]. For instance, the ρ(r) and ∇^2^ρ(r) values for the σ-complexes clearly correlate (see Table 2) well with the interaction energies and equilibrium distances. For the π-hole complexes a direct comparison cannot be made because in some complexes the bond path connects the electron rich atom with the N atoms and in others with the C-atom. Taking complexes **6**, **7**, and **9** where the bond path connects the electron rich atom to the C-atom, ρ(r) and ∇^2^ρ(r) values correlate with the interaction energies and equilibrium distances. The values of ρ(r)_πh_ and ρ(r)_σB_ for all three component systems **11**−**30** are included in Table 3 along with their variation with respect to the binary complexes **2**−**10** [Δρ(r)_πh_ and Δρ(r)_σB_]. The Δρ(r)_σB_ values are positive in all ternary complexes apart from **20**, thus indicating a reinforcement of the σ-bonding by the π-hole interaction. It is interesting to point out that the Δρ(r)_σB_ values are in most cases larger than the Δρ(r)_πh_ ones, suggesting that the reinforcement of the σ-bonding by the presence of the π-hole interaction is the dominant effect.

Apart from CO complexes **15** and **25**, the Δρ(r)_πh_ values summarized in Table 3 are positive, thus indicating a reinforcement of the interaction. In **15** and **25** the negative values of Δρ(r)_πh_ are due to a displacement of the CO toward the aromatic system (see Figure 6a for complex **15**) that enlarges the equilibrium distance and, consequently, decreases the value of ρ(r)_πh_ in the ternary complex with respect to the binary one. In Figure 7, we have represented the distribution of critical points (CPs) and bond paths for several complexes. In Figure 7a, we show the distribution of complex **13**. The H-bond is characterized by a bond CP and bond path interconnecting the H and O atoms. The π-hole is also characterized by a bond CP and bond path interconnecting both N-atoms. It can be observed that the Me_3_N moiety is also connected to the nitro group by two additional bond CPs and bond paths that connect the O-atoms of the nitro to two H-atoms of the trimethylamino molecule. This H-bonds are expected to be very weak since the directionality is very poor (C–H···O angle is 109.8⁰). In ternary complex **14** (Figure 7b) the ditopic nitromethane is connected to the aromatic ring by means of three CPs, symmetrically distributed. One CP connects the O-atom of nitromethane to the C-atom in para. The other O-atom of the nitromethane is connected by two bond CPs and bond paths to the two C-atoms in *ortho*. In complex **16**, the halogen bond is characterized by a bond CP interconnecting the I and O atoms (see Figure 7c). Moreover, the nitrate anion is connected to the ring by four CPs, two of them connect the anion to the C-atom in para and the other two connect the anion to the C-atoms in *meta*. In complex **25** (see Figure 7d), the π–hole interaction is characterized by a bond CP and bond path interconnecting the C atom of CO to the N-atom of the nitro group. Finally, in complex **27** the anion is connected to the ring by means of four CPs and bond paths. One F-atom is connected to the C-atom in para and other F-atom is connected to the N-atom of the N-oxide group and the two C-atoms in *ortho*. In Figure 7, we have also indicated in blue the values of ρ(r) at the bond CPs that characterize the σ-bonding and the π-hole interactions. Moreover, we have indicated in red the values of ρ(r) in the respective binary complexes (values from Table 2). For the σ-bonding interactions, it can be observed that the blue values are in all cases greater than the red ones, thus confirming the reinforcement of the interaction in the ternary complex. Apart from complex **25** commented above, the same behavior is observed for the values of ρ(r) at the bond CP that characterizes the π–hole interaction, that is the blue values are larger than the red ones, thus indicating a reinforcement of the π-hole interaction in the ternary complex in agreement with the energetic and geometric results.

## 3. Computational Methods

The energies of all complexes included in this study were computed at the PBE0-D3/def2-TZVP level of theory. The geometries have been fully optimized imposing *C*_s_ symmetry constraints by using the program TURBOMOLE [76]. X-ray coordinates of all optimized complexes are provided in the Appendix A. All complexes and monomers are true minima, as confirmed by frequency analysis. The interaction energy (or binding energy in this work) ΔE, is defined as the energy difference between the optimized complex and the sum of the energies of the optimized monomers. For the calculations we have used the Weigend def2-TZVP [77,78] basis set and the PBE0 [79]–D3 [80] DFT hybrid functional. For I and Ag, the basis set includes scalar-relativistic calculations with effective core potentials (ECPs) [81,82]: Relativistic effects are taken into account using the Dirac–Hartree–Fock ECPs [83]. The MEP (Molecular Electrostatic Potential) calculations have been carried out by means of the Gaussian-09 software [84] at the same level of theory. The AIM method is used to obtain the distribution of critical points (CPs) and bond paths via analysis of the topology of the electron density [85], which has been carried out at the same level of theory using the AIMALL program [86]. Since we have used the uncorrected energies in this study, for the weaker complexes (those complexes where ΔE > –10 kcal mol^−1^) we have examined is the basis set superposition error is important. The differences between the corrected and uncorrected energies are in all cases less than 5% of the uncorrected interaction energy, thus not affecting the results and discussion. For instance, in the weakest complex **10**, the corrected energy is 1.81 kcal mol^−1^ almost identical to the uncorrected one (1.9 kcal mol^−1^).

In complexes in which the π–hole interaction coexists with other noncovalent interaction (HB or XB), we computed the cooperativity energy E_coop_ using Equations (1)–(5):E_coop_ = ΔE(ABC) − ΔE(AB) − ΔE(BC) − ΔE(AC)(1)
ΔE(ABC) = E(ABC) − E(A) − E(B) − E(C) (2)
ΔE(AB) = E(AB) − ΔE(A) − ΔE(B)(3)
ΔE(BC) = E(BC) − ΔE(B) − ΔE(C)(4)
ΔE(AC) = E(AC*) − ΔE(A) − ΔE(B)(5)
where ΔE(ABC), ΔE(AB), and ΔE(BC) terms correspond to the interaction energies of the three component and two component systems. ΔE(AC) is the interaction energy of the electron rich molecule (the lone pair donors in Scheme 1) with the H/X bond donor (CF_3_I or HF) as they stand the ternary complex (denoted as AC*). This equation has been successfully used before to study cooperativity effects in several multicomponent systems [40,41,42].

## 4. Conclusions

The CSD search reported herein provided strong experimental evidence of the ability of the nitro’s π-hole in *p*-nitropyridine-1-oxide to interact with electron rich atoms. In fact, 80% of the X-ray structures available in the database participate in this type of bonding. More importantly, the N-oxide group also participates in a series of σ-bonding interactions, ranging from noncovalent hydrogen and halogen bonding to covalent coordination bonds and triel bonding. The theoretical DFT results reported herein evidence cooperativity effects between the π-hole interaction involving the nitro group and σ-bonding, either covalent or noncovalent. We have evaluated the cooperativity effects energetically by means of cooperativity energies (*E*_coop_) in those complexes where both interactions are noncovalent and by using binary energies in the covalent ones. Remarkably the *p*-nitropyridine-1-oxide is able to communicate the synergetic effect from the anion or lone pair donor through the conjugated π-system to the N–O thus reinforcing the σ-bonding and vice versa, thus explaining their prevalence in X-ray structures. Finally, molecular electrostatic potential calculations evidence that electrostatic effects are important since the positive potential at the π-hole increases when the N-oxide group interacts with Lewis acids.

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
