# Peer review of "Nitropyridine-1-Oxides as Excellent π-Hole Donors: Interplay between σ-Hole (Halogen, Hydrogen, Triel, and Coordination Bonds) and π-Hole Interactions"

_ijms, 2019, doi:10.3390/ijms20143440_

Round 1
Reviewer 1 Report
This paper describes the computational studies for the p-hole interactions of nitropyridine-1-oxide derivatives along with the comparison with the related experimental data observed in the single crystal X-ray structural analyses. The experimental data of the crystal structures is extracted from Cambridge Structural Database, which is analyzed to detect the p-hole interactions of nitropyridine-1-oxide derivatives as the results of the measurements of intermolecular distance. The DFT-based molecular electrostatic potential (MEP) calculations of the nitropyridine-1-oxide derivative are performed to estimate the trend of N-oxide coordination on a p-hole donor capacity. The synergistic effects between p-hole and s-hole interactions are also studied by the MEP surface analyses of binary and ternary systems and the Bader’s quantum theory of atoms in molecules. The paper contents themselves seem to be sufficiently rich and may be suitable for the article in International Journal of Molecular Sciences. However, there are several issues which need to be addressed and are mentioned below.
1) The word “Triel” in title: The word “triel” is typically used for the atoms in Group 13 according to the related literatures, e.g., see (i) Angew. Chem. Int. Ed. 2000, 39, 670; (ii) Inorg. Chem. 2007, 46, 10047; (iii) ChemPhysChem, 2015, 16, 1470. Although the authors use the words for the descriptions concerning BF3-coordinated molecules in the manuscript, the inclusion of the word in the title makes the readership misleading. Since nitropyridine-1-oxide itself excludes the triel bond, the title should be reconsidered and changed accordingly.
2) p2, lines 58-61: Since the authors have already studied the p-hole donor capacity of 4-nitropyridine, the effects of N-oxide introduction to 4-nitropyridine framework on the p-hole donor capacity should be discussed in the manuscript by the comparison with that of your previous work.
3) p5, descriptions for Figure 3: The tendency of p-hole donor capacity at the ring centroid as the additional p-hole should be also described at this point.
4) p6: Table 2 should appear before Figure 4.
5) p6, Table 2: The superscripts (1 and 2) for the footnote in Table 2 look like digits. The authors should use different indicators such as a and b.
6) p6, line 173: The compound name 15 seems to be incorrect. Probably, it is 9.
7) Style and language should be also polished up through the manuscript, because I really found a lot of typos and incorrect grammars. The page in references should be page-to-page.
Author Response
1) The word “Triel” in title: The word “triel” is typically used for the atoms in Group 13 according to the related literatures, e.g., see (i) Angew. Chem. Int. Ed. 2000, 39, 670; (ii) Inorg. Chem. 2007, 46, 10047; (iii) ChemPhysChem, 2015, 16, 1470. Although the authors use the words for the descriptions concerning BF3-coordinated molecules in the manuscript, the inclusion of the word in the title makes the readership misleading. Since nitropyridine-1-oxide itself excludes the triel bond, the title should be reconsidered and changed accordingly.
Reply: We have modified the title trying to clarify the manuscript topic. Nitropyridine-1-oxide participates as electron aceptor in pi-hole interactions and as electron donor (via de N-oxide O-atom) in triel, halogen and hydrogen bonding and coordination bonds.
2) p2, lines 58-61: Since the authors have already studied the p-hole donor capacity of 4-nitropyridine, the effects of N-oxide introduction to 4-nitropyridine framework on the p-hole donor capacity should be discussed in the manuscript by the comparison with that of your previous work.
Reply: Done (see lines 286-292).
3) p5, descriptions for Figure 3: The tendency of p-hole donor capacity at the ring centroid as the additional p-hole should be also described at this point.
Reply: Done (see lines 152-158).
4) p6: Table 2 should appear before Figure 4.
Reply: Done.
5) p6, Table 2: The superscripts (1 and 2) for the footnote in Table 2 look like digits. The authors should use different indicators such as a and b.
Reply: Done.
6) p6, line 173: The compound name 15 seems to be incorrect. Probably, it is 9.
Reply: Fixed, thank you for taking this to our attention.
7) Style and language should be also polished up through the manuscript, because I really found a lot of typos and incorrect grammars. The page in references should be page-to-page.
Reply: Our apologies. We have done our best to improve the manuscript by correcting the typos and wrong sentences. We have provided the final page number in all references where it is possible. Some of them refer to article numbers.
Reviewer 2 Report
The manuscript is a thorough study on the binary and ternary complexes found between nitropyridine-1-oxides, Lewis bases and a wide variety of anions. The methodology selected is adequate and the work is competitively conducted. The discussion is well-written and easy to read and the literature cite is adequate as well.
The authors have make an effort to identify trends and relationships in the data, paying attention to digest and more importantly to identify and explain the outliers.
The search on the CSD is a well-conducted and demonstrate the importance of combination of experimental and computational work. Also, the synergy between pi-hole and s-hole and the s-hole reinforcement due to the presence of the pi-hole are well explained.
I believe that the paper is a nice piece of work which deserves being publish under IJMS as it stands.
Author Response
We thank this reviewer for his/her careful reading of the manuscript and for supporting publication
Reviewer 3 Report
This paper describes non-covalent interaction nitropyridine-1-oxide with various molecules through p-hole. First from CSD database study experimental evidence show this type of non-covalent interaction. Next as computational results MEP values above and below the p-hole of the nitro group are direct evidence for the existence of p-hole. DFT was mainly used as their computational technique, but qualitatively results may not change even if other techniques were used.
The authors used DFT/def2-TZVP level to estimate delta E etc. Some complexes are less than 10 kcal/mol. In those cases, basis set superposition error may be significant to compare with the binding energy. They should consider and re-estimate these values.
After re-considerations the above, I can recommend this paper is accepted to the journal of International journal of molecular sciences.
Other minor things,
Page 7, line 1(line 194): “-0.254 A” should be “-0.254 Angstrom”. The same font should be used with the other cases.
Author Response
Comment 1. The authors used DFT/def2-TZVP level to estimate delta E etc. Some complexes are less than 10 kcal/mol. In those cases, basis set superposition error may be significant to compare with the binding energy. They should consider and re-estimate these values.
Answer 1: We have computed the corrected binding energies and they differ less than 5% with respect to the uncorrected ones (as usual for DFT methods). We have added a paragraph in the Theoretical methods section, lines 358-363 of the revised mansucript, explaining this.
Comment 2: Page 7, line 1(line 194): “-0.254 A” should be “-0.254 Angstrom”. The same font should be used with the other cases.
Answer 2. Fixed